# Oxidative-Signaling in Neural Stem Cell-Mediated Plasticity: Implications for Neurodegenerative Diseases

**DOI:** 10.3390/antiox10071088

**Published:** 2021-07-06

**Authors:** Mafalda Ferreira dos Santos, Catarina Roxo, Susana Solá

**Affiliations:** Research Institute for Medicines (iMed.ULisboa), Faculty of Pharmacy, Universidade de Lisboa, Avenida Professor Gama Pinto, 1649-003 Lisbon, Portugal; mafaldafsantos@campus.ul.pt (M.F.d.S.); catarinaroxo@edu.ulisboa.pt (C.R.)

**Keywords:** antioxidant, degenerative diseases, mitochondrial dysfunction, neural stem cells, oxidative stress, regeneration, secretome

## Abstract

The adult mammalian brain is capable of generating new neurons from existing neural stem cells (NSCs) in a process called adult neurogenesis. This process, which is critical for sustaining cognition and mental health in the mature brain, can be severely hampered with ageing and different neurological disorders. Recently, it is believed that the beneficial effects of NSCs in the injured brain relies not only on their potential to differentiate and integrate into the preexisting network, but also on their secreted molecules. In fact, further insight into adult NSC function is being gained, pointing to these cells as powerful endogenous “factories” that produce and secrete a large range of bioactive molecules with therapeutic properties. Beyond anti-inflammatory, neurogenic and neurotrophic effects, NSC-derived secretome has antioxidant proprieties that prevent mitochondrial dysfunction and rescue recipient cells from oxidative damage. This is particularly important in neurodegenerative contexts, where oxidative stress and mitochondrial dysfunction play a significant role. In this review, we discuss the current knowledge and the therapeutic opportunities of NSC secretome for neurodegenerative diseases with a particular focus on mitochondria and its oxidative state.

## 1. Introduction

Neural stem cells (NSCs) are multipotent cells present in the developing and adult mammalian central nervous system (CNS) [1]. The term “neural stem cell” describes cells that have the capacity for self-renewal; can give rise to more differentiated cells by asymmetric cell division; and therefore, can generate neural tissue [2]. In fact, NSCs are tripotent, meaning that, through differentiation, they originate neurons, astrocytes and oligodendrocytes [3,4].

The neurogenic niches are defined as the CNS regions where stem cells are retained after embryonic development for production of new cells [5]. These niches provide the physical support to host and anchor stem cells while also suppling factors to maintain and regulate NSC activity [6]. A few years ago, it was believed that neurogenesis was restricted to the embryonic life, and that new neurons would not be generated after birth. In 1965, Altman and Das were the firsts to suggest that neurogenesis also occurs beyond development in the adult mammalian brain [7]. Currently, two main neurogenic regions are identified in the adult mammalian: the subgranular zone (SGZ) in the dentate gyrus (DG) of the hippocampus, which gives rise to granule cells that encodes information related to learning and memory [8]; and the subventricular zone (SVZ) of lateral ventricles (LV), where newborn neurons can migrate along the rostral migratory stream, connecting the olfactory bulb, and differentiate into mature neurons to process olfactory input [9]. Interestingly, studies have suggested that SVZ neurogenesis contributes to social behavior such as mate recognition [10,11] while continued SGZ neurogenesis plays a role in the maintenance of hippocampus plasticity, such as hippocampus-dependent learning, memory and mood function [12]. Nevertheless, adult neurogenesis may also occur in *substancia nigra*, striatum, amygdala and neocortex [13].

Adult NSCs can shuttle between quiescent and active states by exiting and entering the cell cycle, respectively. Once activated, NSCs choose between different modes of division. Of note, the asymmetric division is also self-renewing and originates an NSC and a progenitor, while the symmetric division yields either two NSCs (self-renewing) or two progenitors (not self-renewing). Progenitors, in turn, could be fate restricted, meaning that they can only differentiate into a particular cell type, or be multipotent, making a fate choice before differentiation [3]. Therefore, based on their plasticity, NSCs have been proposed for toxicity studies, tissue engineering and repair, drug target validation and testing, personalized medicine and cellular therapies [14]. In fact, considering their proprieties, such as their high levels of proliferation, low immunogenic reactivity and differentiation potential [15], NSCs present an enormous therapeutic value. In regenerative cellular therapies, for example, NSCs can be transplanted directly into damaged tissues [16]. After transplantation, they could further differentiate into different cells, reconnect with local circuits [17] and release trophic factors, such as growth factors, stimulating antioxidants and immunomodulators [16]. Curiously, until recently, most studies have been focused on the broad plasticity of stem cells and their ability to act as tissue-specific progenitors to substitute cell loss, repair tissue damage and restore function locally. Nonetheless, recent literature has also brought attention to the array of bioactive molecules produced by these cells, such as chemokines, cytokines, growth factors and angiogenic factors, known as the “secretome”, for its important role in the regulation of numerous physiological processes in recipient cells [18]. Notably, the regenerative potential of NSC-derived secretome is often backed on their antioxidant role.

## 2. The Mitochondrial Oxidative State in Neural Stem Cell Fate

Mitochondria are intracellular organelles with two separate and functionally distinct outer and inner membranes [19]; they have their own small circular DNA, mitochondrial DNA (mtDNA), which are molecules of approximately 16.6 kb [20]. Mitochondria participate in a wide range of cellular process, such as energy generation, calcium signaling, redox homeostasis, differentiation, proliferation and apoptosis [21]. Importantly, they are very dynamic organelles that continuously undergo fusion and fission events, processes catalyzed by a family of GTPases [22]. These dynamic events are essential to maintain mitochondrial health, since they allow the elimination of damaged mitochondrial proteins, promote mtDNA complementation, regulate metabolic homeostasis and enable ultrastructural cristae changes in response to altered energy demands [23].

Mitochondrial fusion requires a coordinated process of outer and inner membrane fusion. Inner membrane fusion is mediated by the GTPase Optic atrophy 1 (OPA1) whereas outer membrane fusion requires the formation of mitofusins 1 and 2 (MFN1 and MFN2) oligomeric complexes. Mitochondrial fission, in turn, is mediated by the GTPase dynamin related protein 1 (DRP1) and other factors, such as mitochondrial fission 1 protein (Fis1), mitochondrial fission factor (MFF) and mitochondria dynamic proteins of 49 and 51 kDa (MiD49 and MiD51). These proteins promote the division of mitochondria at the inner and outer membranes [22,24].

Curiously, NSCs have predominately a glycolytic activity. The glycolytic nature is due to their cellular demands, hypoxic microenvironment and is also necessary for the maintenance of stemness [25]. In fact, in NSCs, mitochondria are generally characterized as perinuclear-localized, fragmented morphology and with fewer cristae. At this point, mitochondria are in an immature state, in which oxidative phosphorylation (OXPHOS), ATP and reactive oxygen species (ROS) levels are low [21]. At early stages of neural differentiation, an increase in mitochondrial biogenesis and maturation of mitochondrial morphology occurs, easily observed through mitochondrial elongation and increased cristae number [22]. These morphological adaptations are therefore necessary for the shift in energy metabolism from glycolysis to OXPHOS in response to higher cellular energetic demands of mature neural cells. Thus, mitochondrial ATP, OXPHOS and ROS levels increase in differentiating cells, and a sharp reduction in glycolysis-related proteins occurs. In addition, there is a switch from pyruvate kinase (PK)-M2 to its constitutively active isoform PK-M1 with an upregulation of OXPHOS related genes [25]. Besides the enhanced mitochondrial respiratory capacity, differentiating NSCs also exhibit increased mtDNA copy number. For that reason, the maintenance of mtDNA integrity is imperative for the proper maturation of mitochondria during the process of neural differentiation. In fact, the accumulation of mtDNA damage results in a failure of NSCs to properly differentiate [22].

At the transcriptional level, it was observed that undifferentiated cells, including embryonic stem cells (ESCs) and induced pluripotent stem cells (iPSCs), have elevated mRNA levels of the mitofission gene *Drp1*. Conversely, differentiated cells have increased the abundance of mitofusion *Mfn1* and *Mfn2* mRNAs and protein levels of MFN1, MFN2 and OPA1 [21]. Surprisingly, in NSCs, the progression to a more committed fate is initially accompanied by the transition of the mitochondria to a fragmented state [26]. Indeed, recent findings suggest that, when NSCs stop self-renewal and differentiate into postmitotic neurons, the newborn neurons have high levels of mitochondria fission. In turn, daughter cells destined to self-renew undergo mitochondrial fusion [27].

Curiously, microRNAs are also involved in the energy homeostasis pathway, and also regulate NSC fate. The microRNA-137, for instance, is responsible for the induction of the master regulator of the mitochondrial biogenesis, the peroxisome proliferator-activated receptor and coativator 1α (PGC1α)-independent mitochondrial biogenesis. Indeed, miRNA-137 indirectly upregulates both the nuclear factor erythroid 2-related factor 2 (Nrf2) and the mitochondrial transcription factor A (TFAM). Nrf2 is a master redox regulator, which is subsequently translocated to the nucleus to regulate transcription of differentiation- or self-renewal-related genes [22], while TFAM enhances mitochondrial replication and biogenesis, following by the activation of OXPHOS machinery. This microRNA also promotes morphological changes in the mitochondria by modulating mitochondrial fusion and fission proteins [28].

Interestingly, a nuclear retrograde signaling is also observed throughout the regulatory mechanisms of NSC fate, in which transcription of self-renewal- and differentiation-related genes are regulated by mitochondrial signals. For example, a physiological or moderate elevation of mitochondrial ROS serves as a signaling mechanism to stabilize Nrf2 levels. Further, Nrf2 can strongly influence the NSC fate and function through the reduction of ROS to the benefit of NSC survival and neurogenesis [29].

Previous studies also showed that mitochondrial translocation of p53 alleviates mitochondrial damage and apoptosis-related events in the context of neural differentiation [30]; and that an endogenous bile acid, tauroursodeoxycholic acid (TUDCA), enhances NSC proliferation, self-renewal and neuronal conversion by restoring mitochondrial integrity, increasing ATP levels and reducing mitochondrial ROS levels in NSCs [31]. Curiously, it has been later demonstrated that this NSC proliferation-inducing molecule increases cell division by inducing de novo lipogenesis and a metabolic shift from fatty acids to glucose catabolism that facilitates NSC cell cycle-associated histone 3 (H3) acetylation. Therefore, it has become evident that metabolic status is indeed pivotal to determine NSC fate and activity [32].

Recent evidence has also shown that neural function, including adult neurogenesis, is tuned by the individual diet and respective gut microbiome (GM), where host metabolism plays a key role [33]. In fact, short chain fatty acids (SCFAs), major products of dietary fiber fermentation by GM in the large intestine, enter in the systemic circulation and directly affect mitochondrial metabolism of other tissues, such as brain [34]. In this regard, it has been recently revealed that these types of microbial metabolites, increased by specific diets, trigger mitochondrial biogenesis in NSCs, induce premature neurogenic differentiation through a ROS- and extracellular signal-regulated kinases 1/2-dependent manner and lead to a rapid NSC pool depletion [35].

## 3. The Mitochondrial Oxidative State in Neurodegenerative Diseases

Cellular oxidative stress is caused by an imbalance between production of ROS and the enzymatic or non-enzymatic detoxification of reactive species [36,37,38]. When there is an overproduction or a loss detoxification of ROS, several cellular structures, including membranes, lipids, proteins, lipoproteins and DNA, are negatively affected [36]. These ROS-mediated effects lead to structural changes in these macromolecules that ultimately also impair their function and activity [37]. As an example, lipid peroxidation can trigger mitochondrial swelling and uncoupling electron transport from ATP synthesis. Moreover, lipid hydroperoxides are involved in damage of specific mitochondrial proteins and transport systems [39]. Noteworthy, the brain is one of the most metabolically active organs in the human body, being particularly prone to oxidative damage. In fact, the high rate of oxidative metabolic activity in the brain causes necessarily an intense production of their oxidative byproducts. Therefore, if the antioxidative defense system in the brain is not robust enough to scavenge ROS, these high reactive molecules end up by reacting with neurotransmitters, lipids, proteins and even with nucleotide molecules, such as RNA and DNA, in all type of the CNS cells [40]. In agreement with this idea, it is known that the vast majority of neurological disorders are somehow related with disturbed mitochondrial energetic metabolism.

The endogenous production of ROS occurs mainly in mitochondria, where most ROS are generated in the electron transport chain [41,42]. In addition, while mitochondrial dysfunction impairs energy production and calcium storage capacity, it is also responsible for increasing ROS production. The excess ROS, in turn, will further damage mitochondria, modify their metabolic enzymes and lead to its further dysfunction [37,43]. In this sense, increased mitochondrial ROS, and consequently, mitochondrial oxidative stress, can be the cause and consequence of mitochondrial dysfunction (Figure 1) [44].

Neurodegenerative diseases are known as the most prevalent disorders with high burden to the patients, their families and society [45]. The prevalence of these types of illnesses is increasing, owing, in part, to extensions in lifespan [46]. They are characterized by a progressive degeneration of specific subsets of neurons, such as dopaminergic, cholinergic or motor neurons, resulting in a precise pattern of nervous system dysfunction [47]. The role of mitochondrial dysfunction and excessive levels of ROS in the neurodegeneration process has thus been largely described throughout the last years [37]. Indeed, along with individual metabolic alterations, abnormal mitochondrial function, motility and biogenesis are found in the early stages of the majority of neurodegenerative diseases. This can result from mitochondrial damage, mutations in mitochondria-interacting nuclear proteins and compromised mtDNA integrity. Alzheimer’s disease (AD) and Parkinson’s disease (PD) are two examples of disorders associated with mitochondria and oxidative stress that we will further discuss [48].

### 3.1. Alzheimer’s Disease and Mitochondria Dysfunction

AD is the most common cause of dementia, characterized by synaptic degeneration and neuronal loss throughout the brain, particularly in the basal forebrain, amygdala, hippocampus and cortical areas [49]. The two most common hallmarks of AD brains are the presence of amyloid β (Aβ) peptides and neurofibrillary tangles (NFTs) of hyperphosphorylated tau protein [50]. Curiously, mitochondria isolated from AD patients were shown to exhibit structural changes, such as reduced length and overall size. In addition, AD patient-derived mitochondria present metabolic changes, including abnormal redox potentials and weakened mitochondrial metabolic potential. These changes are accompanied by other significant functional impairments, such as the inability to maintain adequate levels of calcium and electrical potential, ATP production and low levels of ROS [51].

As the name suggests, Aβ peptide is derived from amyloid precursor protein (APP) through sequential enzymatic cleavages promoted by β- and γ-secretases [52,53,54]. APP mutations occur in familial AD (FAD), whose transmission is autosomal dominant. Specifically, the *APPswe* mutation leads to a 5–10-fold increase in the production of Aβ_40_ and Aβ_42_, while the *APParc* mutation induces the production of Aβ with greater propensity for the formation of protofibril [52]. Curiously, APP has already been identified in the mitochondria, preventing the import of proteins. Indeed, it thus appears that APP is retained in the mitochondria and forms a complex with the translocases of the internal and external mitochondrial membrane. This transmembrane arrest of APP was shown to induce mitochondrial dysfunction, such as disruption of mitochondrial membrane potential and ATP synthesis, but also to reduce cytochrome oxidase activity [53]. Furthermore, even when APP mitochondrial localization is attenuated, mitochondrial disfunction is still occurring in multiple ways, namely, by its interference with mitochondrial fission and fusion dynamics [55]. On the other hand, autosomal dominant mutations in presenilin genes are responsible for ~90% of FAD cases [56]. Presenilin (PS) 1 and 2 form the catalytic core of γ-secretase [53,57]. One of the possible causes for the involvement of PS in FAD is the association of PSs with intracellular calcium stores of the endoplasmic reticulum (ER) and the mitochondria, a γ-secretase-independent process [58]. In addition, FAD-PS2 mutants reduce the levels of ER and medial-Golgi apparatus luminal calcium. In fact, it is well established that calcium regulates cellular bioenergetics by allosterically activating key metabolic enzymes and shuttles or indirectly modulating signaling cascades. For example, the correct transport of Ca^2+^ between ER and mitochondria compartment enhances Kreb Cycle and sustains ATP production. Thus, considering this effect on ER-calcium content, the mutants dampen mitochondrial calcium rises, impairing ATP production [59]. Subsequently, the inability of mitochondria to produce enough energy to supply massive energy demands of neurons leads to cell death [56]. Moreover, low levels of cellular ATP cause neuronal incapacity to face high calcium workflow, easily leading to glutamate-induced excitotoxicity, and further sustaining mitochondrial calcium overload, ROS production and neurodegeneration [60]. Therefore, proteins involved in familial early onset forms of AD are major contributors of mitochondrial dysfunction and redox imbalance, which in turn trigger neurodegeneration.

Regarding the sporadic forms of AD, which accounts for >95% of all cases, oxidative stress, Aβ and NFTs are also involved in a vicious cycle. Oxidative stress induces Aβ aggregation and facilitates the phosphorylation and polymerization of tau while protein deposition, in turn, increases oxidative stress [38]. Interestingly, it has been shown that oxidative stress may activate signaling pathways that alter APP and tau processing. High levels of ROS increase β-secretase expression through activation of c-Jun amino-terminal kinase and trigger aberrant phosphorylation and polymerization of tau via activation of p38-activated protein kinases [61] and fatty acid oxidation, respectively [38]. Further, Aβ directly interacts with mitochondria, resulting in impaired activity of complexes III and IV, reduced ATP production, mitochondrial calcium overload, disturbance of mitochondrial fusion and fission and increased generation of ROS [43,62]. In addition, expression of tau protein increases cellular vulnerability to oxidative stress, which in turn has been related to depletion of peroxisomes [38]. On the other hand, different studies have suggested a mechanistic link between AD and glutamate excitotoxicity. Aβ induces a partial blockage of N-methyl-D-aspartate (NMDA) receptor, leading to an increase of presynaptic glutamate release and consequently to an increase of glutamate concentration in synaptic cleft [63]. Curiously, mitochondrial dysfunction has been indicated as one of the primary events in glutamate excitotoxicity. Again, a vicious cycle occurs since oxidative stress triggered by mitochondrial dysfunction promotes glutamate release [38] and increased levels of ROS found in AD brains stimulate pro-inflammatory gene transcription. Microglia respond to neuroinflammation and also generate large amounts of ROS [64]. Therefore, activated microglia also function as an amplified source of ROS in AD [38].

At last, mtDNA molecules do not have histone protection and, being located close to the place where free radicals are produced [37,43,64], are more prone to oxidative damage and mutations [43]. Curiously, the transfer of mtDNA from patients with AD to cell lines without mtDNA induces deficiency of respiratory enzymes such as those observed in AD tissues. This suggests that the mitochondrial deficit is carried, at least in part, by mtDNA abnormalities [65]. In fact, it has been proposed that mtDNA deletions, which accumulate with age, may be responsible for complex IV deficiency found in AD [66]. However, the specific role of mtDNA changes in the pathogenesis of AD has not been clarified yet [44].

### 3.2. Parkinson’s Disease and Mitochondrial Dysfunction

Parkinson’s disease (PD) is a neurological disorder characterized by classical motor features of parkinsonism, such as tremor, rigidity and bradykinesia, and well as accumulation of Lewy bodies in the brain, loss of dopaminergic neurons in the *substantia nigra* and decreased dopamine levels [67]. Curiously, mitochondrial dysfunction was first associated with PD when recreational drug users were exposed to 1-methyl-4-phenyl-1,2,3,6-tetrahydropyridine (MPTP), since its metabolite 1-methyl-4-phenylpyridinium (MPP^+^) inhibits complex I of the mitochondrial electron transport chain [61,66]. Indeed, the most common characteristic of mitochondrial dysfunction in PD patients is the reduced activity of complex I, which in turn results in a major source of ROS levels. Of note, the mitochondrial complex I deficiency is not only observed in *substancia nigra* of PD patients, but also in the frontal cortex, fibroblasts, skeletal muscle, lymphocytes and blood platelets of these individuals [43,68]. Importantly, recent genetic discoveries have supported the relevance of mitochondria in PD pathology, once multiple PD-associated genes encode proteins that are relevant to mitochondrial homeostasis [69], such as genes encoding for presynaptic protein α-synuclein, E3 ubiquitin ligase Parkin, PTEN-induced putative kinase 1 (PINK1), protein deglycase DJ-1, leucine-rich repeat kinase 2 (LRRK2), ATPase 13A2 and vacuolar protein sorting-associated protein 35 [69,70].

In PD, α-synuclein accumulates in insoluble fibrils and is the major component of Lewy bodies and Lewy neurites [71]. Mice with A53T α-synuclein mutations present increased mtDNA damage and mitophagy, indicating that α-synuclein disrupts mitochondrial homeostasis [72]. The α-synuclein can also induce loss of endoplasmic reticulum-mitochondria contacts, disrupting Ca^2+^ exchange between the two organelles and ATP production [73]. Additionally, it has been revealed that toxicity of α-synuclein is specifically dependent on mitochondrial uptake of Ca^2+^, as well as on electron flow through complex I [74]. However, in addition to complex I, α-synuclein oligomers were shown to have high affinity to the translocase of the outer mitochondria membrane (TOM) 20 peptide receptors, preventing TOM20 and its co-receptor TOM22 binding, inhibiting respiration and increasing ROS production [75]. Further, a link between α-synuclein and the master regulator of mitochondrial biogenesis, PGC-1α, has been also observed. In fact, a downregulation of PGC-1α was already shown in the cell model for α-synuclein oligomerization, A30P α-synuclein transgenic mice and human PD brains. In these studies, the pharmacological activation and overexpression of PGC-1α was shown to reduce α-synuclein oligomerization and neuronal loss [72,76].

In addition, the genes encoding for Parkin and PTEN-induced kinase 1 (PINK1) proteins are associated with an autosomal recessive form of PD [77], and it has been already reported that loss of these proteins induces mitochondrial dysfunction and, consequent, overproduction of ROS [78]. Indeed, both Parkin and PINK1 play a pivotal role in mitophagy, a critical process to eliminate damaged mitochondrial and maintain healthy mitochondria and their functions [69]. Thus, reducing the ability of cells to remove impaired mitochondria, through the loss of Parkin or PINK1, or due to the presence of abnormal proteins, may cause an accumulation of these dysfunctional organelles leading to early onset-PD [79].

Moreover, mutations in the LRRK2 gene are, in turn, the most prevalent cause of late-onset familial PD, also occurring in sporadic forms of the disease. This mitophagy-related protein is a key mitochondrial regulator [80,81] and mutations of LRRK2 lead to oxidative stress, abnormal mitochondrial fission and fusion, perturbations of calcium homeostasis and impairments in mitophagy and mitochondrial trafficking [82].

On the other hand, the *substancia nigra* is highly exposed to ROS due to dopamine metabolism, high concentrations of iron and reduced levels of antioxidants in this region. Curiously, dopamine is an unstable molecule that undergoes auto-oxidation to form quinones and H_2_O_2_ [83]. Dopamine quinone can suffer intramolecular cyclization, and, consequently, leads to mitochondrial dysfunction, protein degradation, α-synuclein oligomerization and originates superoxide radical [84]. Moreover, the transport and storage of dopamine contributes to elevated ROS production, while some evidence suggests that the synthesis of total GSH is considerably affected in PD patients [85]. Dopamine quinone can react with sulfhydryl groups of the cysteine in proteins, particularly glutathione (GSH), resulting in lower GSH levels [78]. In this way, GSH cannot scavenge ROS and, consequently, contributes to the increase of oxidative stress.

Chronic neuroinflammation has been also reported in PD. Some authors have demonstrated that activated microglia and high levels of pro-inflammatory cytokines are present in the brain with PD [38,68]. Since inflammation is associated with a large release of ROS, oxidative stress can also be exacerbated in PD patients in this way [38]. Importantly, neuromelanin released from dying dopaminergic neurons activates the microglia, further increasing the sensitivity of these neurons to oxidative stress-mediated cell death [68]. At last, pathogenic mtDNA mutations are also associated with PD. *Substancia nigra* neurons from autopsies of normal elderly people and PD patients harbor high levels of mutated mtDNA with large-scale deletions that cause mitochondrial dysfunction. Indeed, mitochondrial disease patients with mutations in polymerase γ, the polymerase responsible for mtDNA replication, excessively accumulate mtDNA mutations and present an increased risk of developing PD [79]. In agreement, mutations in mtDNA-encoded complex I subunits were shown to cause parkinsonism [86].

## 4. Oxydative-Signaling of NSC-Derived Secretome

Cells communicate with each other by intercellular contacts and paracrine secretion of molecules [4]. Notably, several studies showed that stem cells trigger tissue repair due to their ability to secrete molecules with a beneficial impact on the damaged tissue, rather than to their capacity to differentiate into the required differentiated cells [87]. These secreted factors are referred to as cell secretome and include growth factors, cytokines, chemokines and exosomes [4,88] found in the cell culture medium (conditioned medium). The soluble factors and vesicles secreted by stem cells may therefore act directly by mediating intracellular pathways in injured cells, or indirectly, by inducing the secretion of functionally active products from adjacent tissues [87]. More importantly, despite being implicated in repair, restoration and regeneration of injured tissues [4], stem cell secretome has the advantage of being cell-free, with no need of donor–recipient matching [89] and without tumorigenic potential [12]. In addition, secretome can be easily packaged, freeze-dried and transported [89], turning its manipulation even more attractive.

Without exception, NSCs appear to also function as local “factories” capable of producing and secreting a wide range of molecules [90] that target other cells, including their own producing cells. Thus, the paracrine hypothesis proposes that NSC grafts act as a natural source of potent biologics capable of modulating and promoting the restoration of several key functions in the CNS tissue following acute or chronic tissue damage [90]. Unveiling the predictive potential of NSC secretome will certainly help to foster novel therapeutic strategies to many neurological diseases prone to benefit from NSC-mediated neuroplasticity.

### 4.1. Antioxidative Potential of Delivered Neurotrophic Factors

Different studies have demonstrated that NSCs are able to increase the survival and regeneration of endogenous neurons by secreting important neurotrophic factors capable of modulating the CNS development and function. These include the growth factors, such as the nerve growth factor (NGF), the brain-derived neurotrophic factor (BDNF), the neurotrophin-3 (NT3), the glial-cell-line-derived neurotrophic factor (GDNF), the ciliary neurotrophic factor (CNTF), the vascular endothelial growth factor (VEGF) and insulin growth factors (IGF)-I and II [47,90].

Since the 1990s, it is known that mutation in superoxide dismutase (SOD), a key antioxidant enzyme, can originate motor neuron degeneration [91]. In fact, animals with mutant SOD were shown to present progressive paralysis. Surprisingly, Nizzardo et al. treated mutant SOD1 mice, associated with amyotrophic lateral sclerosis phenotype, with iPSC-derived NSCs, resulting in an improved neuromuscular function and a significant increase in motor neuron survival. Among the causes of the beneficial effects, the authors highlight the production of neurotrophic factors by iPSC-derived NSCs [92]. In a way, they suggest that NSC-released neurotrophins, including GDNF, BDNF and NT3, were responsible for rescuing mutant SOD1-induced degeneration in neighboring cells.

It has been described that neurotrophic factors act in major antioxidant defense pathways, rescuing cells from oxidative damage (Figure 2). The NGF, for example, protects cells against oxidative stress, including neurons, by preventing ROS accumulation via phosphatidylinositol 3-kinase (PI3K)/Akt1 and promoting upregulation of the heme oxygenase-1 (HO-1) [93]. The induction of HO-1 expression and its activity has been pointed out to provide cell resistance to oxidative damage [94]. On the other hand, NGF increases the levels and activities of several proteins involved in antioxidant defense systems, including catalase, superoxide dismutase (SOD), glutathione reductase (GSSG-R) and glutathione peroxidase (GSHPx) [95] while also preventing mitochondrial dysfunction by maintaining Ca^2+^ homeostasis and mitochondrial transmembrane potential [96]. Surprisingly, the TrkA and p75 neurotrophin receptors were already shown to be localized in the inner mitochondrial membrane of several differentiated tissues, including in CNS, providing new insights into the direct role of NGF at the mitochondrial level [97].

BDNF is another neurotrophic factor shown to augment activities of antioxidant enzymes in cortical neurons, including SODs, GSHPx and GSSG-R [95]. Recently, it was revealed that BDNF also induces the expression of sulfiredoxin, an ATP-dependent antioxidant enzyme. This process involves the phosphorylation of ERK1/2 with subsequent induction of c-Jun. Further, the BDNF-induced nitric oxide (NO)/cyclic guanosine monophosphate (cGMP)/protein kinase (PK) G pathway contributes to the activation of the transcription factor nuclear factor (NF)-κB. In fact, the exposure to BDNF in neurons induces the formation of a PKG-1/NF-κB complex that drives the expression of sestrin2 and attenuates cellular ROS contents [98]. In addition, BDNF upregulates mitochondrial uncoupling protein 2 (UCP2) and restores mitochondrial electron coupling capacity [98]. Finally, it induces accumulation of phosphorylated cAMP response element (pCREB) in the mitochondrial inner membrane and matrix to help the synthesis of complex V [99]. Thus, BDNF improves respiratory coupling in mitochondria [100].

Regarding NT3, it has been revealed that this neurotrophin is able to reduce ROS levels in adult myenteric neurons [101]. However, NT3 does not promote the expression of antioxidant enzymes [102]. Indeed, the antioxidant effect of NT3 might be due to its effect in restoring the mitochondrial inner membrane potential, and subsequently, in preventing mitochondrial dysfunction [103]. Curiously, the glial neurotrophic factor GDNF significantly increases GSHPx, SOD and catalase activities [104,105], while the stimulation of the endogenous receptor tyrosine kinase Ret by GDNF is capable of restoring complex I activity, respiration and ATP production in *Pink1* mutant cells [106].

Respecting CNTF, it inhibits oxygen glucose deprivation/re-oxygenation (OGDR)-induced ROS production by activating the Akt-Nrf2 signaling pathway in myocardial cells [107]. Based on the fact that Nrf2 is pivotal for the transcription and expression of important antioxidant enzymes [108], it is possible that this signaling pathway might also be involved in CNTF-induced upregulation of SOD2 [109]. Interestingly, CNTF was also revealed to control oxidative stress through the activation of the janus tyrosine kinase 2 (JAK2)/signal transducer and the activator of transcription 3 (STAT3) pathway [110], reversing aberrant mitochondrial bioenergetics [111]. In turn, VEGF induces expression of SOD2, by a NADPH oxidase-dependent mechanism. VEGF activates protein kinase C (PKC) that subsequently phosphorylates and degrades IκB, trigging nuclear translocation NF-κB and inducing SOD2 expression [112].

In the brain cortex and hippocampus, IGF-I also improves antioxidant enzymes activities, such as SOD, catalases and GSPHx [113], being capable of restoring all parameters of mitochondrial dysfunction, such as depletion of membrane potential, increased proton leak rates and intramitochondrial free radical production [39,114]. At last, IGF-II regulates the synthesis and/or activity of SOD, restores the mitochondrial cytochrome *c* oxidase (COX) activity and the mitochondrial membrane potential, decreasing ROS production and rescuing the antioxidant status in adult cortical neuronal cultures [115]. This neuroprotection appears to be reached through the interaction of IGF-II with its specific IGF-II receptor [116]. Therefore, a wide range of soluble molecules secreted by NSCs are functionally involved with metabolic processes and redox signaling, which take part in neuron survival [47].

### 4.2. Antioxidative Potential of Delivered Extracellular Vesicles and Tunneling Nanotubes

Beyond soluble factors, the secretome of NSCs also contains extracellular vesicles (EVs). These vesicles can be considered as paracrine or endocrine signaling vehicles, since they are able to transport defined signaling molecules to distant target cells. Interestingly, EVs are involved in the bidirectional exchange of information between stem cells and target cells, a crucial process for brain repair after injury [4]. Exosomes are small membrane vesicles with ~30–150 nm that represent one sub-type of the EVs [117,118]. They are composed of cellular membranes with associated membrane proteins, surrounding a core containing molecules, such as proteins, lipids, miRNAs and mRNAs [47,117]. The formation of exosomes occurs in multivesicular endosomes and includes several important steps illustrated in Figure 3. Of note, the same cell may release different subpopulations of exosomes with distinct cargo compositions, inducing distinct effects on target cells [12].

The most frequently identified proteins in exosomes include proteins involved in multivesicular body biogenesis; membrane associated proteins (lipid rafts proteins, GTPases and other membrane trafficking proteins); transmembrane proteins (targeting/adhesion molecules, tetraspanins CD9, CD63, CD81, membrane fusion proteins); cytoskeletal proteins (actin, syntenin, moesin); signal transduction proteins (annexin); chaperones (Hspa8, Hsp90); and, importantly, metabolic enzymes (glyceraldehyde-3-phosphate dehydrogenase, lactate dehydrogenase A, phosphoglycerate kinase 1, pyruvate kinase, aldolase) [119].

Recently, the protein signatures of EVs from iPSC-derived NSCs were assessed and one of the most abundant proteins found in these vesicles was the hemopexin (Hpx) [120]. Curiously, Hpx is a heme scavenger, protecting the brain from heme-mediated damage [121]. Indeed, heme is a highly toxic molecule due to its ability to intercalate with lipid membranes and to produce hydroxyl radicals through the release of ferrous iron from the heme catabolism [122,123]. Notably, the high expression of Hpx in nerve cells exposed to blood clot reduces lipid peroxidation and increases the enzymatic activities of GSH and SOD [124]. Therefore, Hpx plays an important role in the antioxidant protection, corroborating the idea that NSC-EVs are a source of proteins involved in antioxidant pathways.

Nevertheless, besides proteins, exosomes are also enriched in cholesterol, ceramide, phosphatidylserine, phosphatidylinositol, sphingomyelin, phosphatidylcholine [12] and, importantly, in miRNAs [119] and mRNAs, which can silence gene expression or be translated in recipient cells [117], respectively. Thus, all these components can modify the phenotype and/or the physiology of the target cell, modulating relevant cellular processes as proliferation, differentiation and survival [119]. For example, it has already been described that several steps of adult neurogenesis are mediated or regulated by miRNAs, such as miRNA-let7b and miRNA-9, which regulate proliferation and differentiation of NSCs [119,125]. Further, miRNA-124, miRNA-128 and miRNA-137 might act cooperatively and synergistically to promote neuronal differentiation of NSCs by targeting overlapping gene sets containing a highly interconnected transcription factor network [119]. Notably, miRNA-21a was found in NSC-derived exosomes and promotes the generation of neurons by activating Akt and Wnt signaling pathways [126], while endogenous hypothalamic NSCs were shown to control the ageing process through the release of exosomal microRNAs in the cerebrospinal fluid [127]. In this last study, the authors revealed that a certain pool of exosomal miRNAs declined during ageing, whereas central treatment with healthy hypothalamic NSC-secreted exosomes led to the slowing of this process. In this sense, nucleic acids transported within exosomes act as regulators of neurogenesis, being relevant for neural regeneration, neuroprotection, neural plasticity and also ageing control [12].

Surprisingly, mitochondrial proteins, mtDNA and full functional mitochondria were also shown to be transferred between mammalian cells, through EVs [128]. Some studies have already demonstrated that this transfer process occurs in mesenchymal stem cells (MSCs). For example, the mitochondria transfer via MSC-derived EVs is responsible for enhancing phagocytosis and anti-inflammatory effects on macrophages by promoting oxidative phosphorylation [129]. In addition, microvesicles from MSCs were shown to transfer partially depolarized mitochondria to macrophages, increasing bioenergetics in these cells in response to oxidative stress [130]. Further, bone marrow-derived MSCs release mitochondria-containing microvesicles that are engulfed by pulmonary epithelium. This transfer was shown to increase alveolar ATP [131]. On the other hand, it has been described that MSC exosomes express functional respiratory complexes I, IV and V, which are capable of consuming oxygen. These exosomes exhibit an aerobic respiratory capacity independent of the whole mitochondria that contributes to their ability to rescue cell bioenergetics [132]. More importantly, the traffic of functional mitochondria was also recently shown to occur in NSCs via extracellular vesicles. For the first time, Peruzzotti-Jametti et al. have demonstrated that transfer of mitochondria from NSC-derived EVs rescues mitochondrial function and cell survival in mtDNA-deficient target cells [133].

Furthermore, mitochondria transfer can also occur through tunneling nanotubes (TNTs) from stem cells to recipient cells (Figure 4). TNTs are nanotubular actin-based structures with a small diameter, ranging from 50 to 200 nm and a length of up to 100 μm. These nanostructures allow the active transport of cargo and mitochondria between two or more cells. The donor cell extends a filopodia-like protrusion toward the recipient cell that provides plasma membrane continuity, thus forming a strait connected bridge between cells that does not touch the substrate [134,135,136]. Notably, it has been demonstrated that neurons are able to establish intercellular contacts with bone marrow mesenchymal multipotent stromal cells (MMSCs) via TNTs, sharing their cytosolic contents, while MMSCs transfer their mitochondria to neurons or astrocytes, leading to the restoration of respiration and stimulation of cell proliferation in recipient cells, reducing ischemic damage [137,138]. Moreover, these type of mitochondrial transfer from MSCs was shown to restore aerobic respiration and improve cell survival of human vein endothelial cells previously subjected to in vitro ischemia-reperfusion [139]. Finally, another recent study revealed that mitochondrial transfer from MSCs via TNTs improves neuronal metabolism after oxidant injury in vitro [140].

However, although it is possible that NSCs also act as potential mitochondrial donors through this TNT pathway, there is no clear evidence regarding TNT-mediated mitochondrial transfer in NSCs. This is particularly relevant because the transfer of healthy mitochondria from NSCs to damaged cells represents an important mechanism of endogenous regeneration by these cells [141]. Hence, it is crucial to further explore and modulate all types of mitochondria transfer in these cells to augment the benefits of NSC-based therapies in the central nervous system.

## 5. NSC-Derived Secretome in Neurodegenerative Conditions

Over the last years, the secretome derived from NSCs aroused the interest of investigators, especially regarding its potential use for the treatment of neurodegenerative conditions. In fact, considering all previously described components of NSC secretome, it is easy to understand why. Several studies have evaluated the beneficial and regenerative effects of neurotrophic factors released by NSCs under pathological conditions. For example, in Huntington’s disease (HD), the increasing striatal BDNF by NSCs was shown to ameliorate motor dysfunction and normalize brain weight, arresting the progression of the disease. In addition, the expression of dopamine- and cAMP-regulated phosphoprotein-32 (DARPP-32) typically impaired in HD mice appeared to be normalized by BDNF overexpression [142]. Similarly, Wu et al. confirmed that BDNF overexpression enhances the therapeutic potential of NSC transplantation in the transgenic mouse model of AD [47]. The upregulation of BDNF improved the survival of transplanted cells, neuronal fate, neurite complexity, maturation of electrical property and the synaptic density [143].

Other examples include the transplantation of NT3-overexpressing NSCs in a rat stroke model, which improved animal motor functions [144], and a recent study using human NSC lines derived from fetal tissue overexpressing IGF-1. Of note, reduced levels of IGF-1 have been associated with cognitive decline. In this recent study, IGF-1 human NSCs (hNSCs) revealed to be neuroprotective against Aβ toxicity in neurons through its paracrine activity [90].

On the other hand, and as previously mentioned, the NCS-derived secretome also include membrane-enclosed structures. Indeed, EVs present several clinical advantages, such as a low cytotoxic and immunogenicity potential and a high permeability, which in turn facilitates their cross through the blood–brain barrier [145]. Curiously, it has been shown that the application of NSC-derived EVs in a rat model of spinal cord injury reduces lesion extents, neuronal apoptosis, microglia activation and neuroinflammation while it also improves animal functional recovery. Interestingly, other recent studies have been suggesting that NSC-EVs regulate apoptosis and inflammatory processes via autophagy. For example, Rong and co-authors have reported that inhibition of autophagy reverts the effects of NSC-derived EVs in reducing the extent of neuronal apoptosis, microglia activation, neuroinflammation and spinal cord injury in rats [146].

Regarding stroke, it is well known that this pathology induces a secondary neurodegeneration [147]. In this sense, Webb et al. demonstrated that treatment with hNSC-EVs in a mouse stroke model improves cellular, tissue and functional outcomes in middle-aged animals. This treatment significantly decreased neural injury, also having a positive effect on motor function, memory formation and chronic inflammation [148]. Later, the same group evaluated the impact of hNSC-derived EVs in a porcine ischemic stroke model. Consistent with their previously results, NSC-EVs were neuroprotective. Indeed, these animals exhibited a decrease in cerebral lesion volume and decreased brain swelling relative to control animals. Importantly, hNSC-EVs also preserved the integrity of white matter regions, when examined three months after stroke [149]. These data help to corroborate the idea that NSC-sourced EVs have, indeed, anti-inflammatory, neurogenic and neurotrophic effects. On the other hand, cerebral ischemia reperfusion injury is a common feature of ischemic stroke [150]. In this regard, hNSC-EVs were demonstrated to be effective in reducing apoptosis and upregulating the expression of SOD1, catalase and GSHPx1 in neuronal cells after hypoxia/reperfusion injury. The investigators speculate that these EVs might promote Nrf2 nuclear translocation, which would explain the increased expression of antioxidants. Moreover, NSC-sourced EVs promoted neuronal axon elongation, thus favoring nerve regeneration and the repair of dysfunctional nerves [151].

Considering AD pathological context, the injection of NSC-derived EVs into the lateral ventricles of the transgenic mice model of AD revealed a rescue effect of the cognitive impairments. This procedure enhanced mitochondrial function, SIRT1 activation and the memory-related synaptic proteins, also reducing the expression of oxidative damage markers, pro-inflammatory cytokines and microglial markers. Therefore, although Aβ levels have not been changed, the exposure of NSC-derived EVs appears be a promising strategy to delay the progression of AD [152].

In addition to neurotrophic factors and EVs structures, the secretome, as a whole, has been investigated and revealed promising in neurodegenerative contexts. A previous proteomic study revealed the presence of Rho-guanine nucleotide dissociation inhibitor 1, phosphatidylethanolamine binding protein (PEBP), polyubiquitin, immunophilin FK506 binding protein 12 (FKBP12) and cystatin C in conditioned medium derived from cultured adult hippocampal progenitors [153]. Cystatin C, for example, is able to partially rescue lesions induced by 6-hydroxydopamine in nigral dopaminergic neurons [154], while FKBP12 is pointed out as a biomarker for prodromal stages in AD and PD [155]. However, more proteomic and transcriptomic analysis must be performed to further identify relevant molecules underlying the regenerative properties of NSC-derived secretome. Lim et al. have confirmed the neuroprotective effect of NSC-secretome in an in vitro model of HD, showing that it significantly reduces Huntingtin aggregates [156]. Further, the secretome derived from adult NSC was shown to protect hippocampal neurons from NMDA-induced excitotoxicity. This was shown to be mediated by a novel neuroprotective peptide, pentitin, a putative breakdown product of the insulin B chain [157]. Another study investigated the effects of hNSCs-derived secretome in a rat model of PD. The injection of secretome from hNSCs revealed a significant increase in dopaminergic neurons survival and functional motor activity, when compared to either the control group or the group in which the animals were transplanted with hNPSCs. More importantly, the proteomic characterization of this secretome revealed that these cells have the ability to secrete a wide range of important molecules with neuroregulatory actions, which confirms the observed effects [158].

Jia et al., in turn, treated SH-SY5Y cells with Aβ, to produce an in vitro model of AD. These cells were treated with NSC-derived secretome which was shown to be very effective in rescuing cell viability and apoptosis. Indeed, NSC-secretome was capable of decreasing ROS content, restoring Bcl-2 expression and attenuating the levels of Bax, cytochrome *c* release and caspase activation induced by Aβ. Additionally, the decrease in mitochondrial membrane potential was shown to be reversed by the secretome derived from NSCs. In fact, this study showed that NSC-secretome has a protective effect on the damage induced by Aβ peptide, partly due to its protective role on mitochondrial activity [159].

Altogether, these evidences suggest that the paracrine effect of NSCs, mediated by soluble factors and/or EVs, are responsible in a major way for the benefits of NSC-based therapies observed in preclinical models of neurodegenerative conditions.

## 6. Summary and Future Directions

After the discovery of stem cells, many types and classifications have been proposed based on their origin and potency/differentiation potential [160]. An important landmark in the history of stem cells was the identification of NSCs and the recognition that neurogenesis also occurs in specific areas of the adult brain. In fact, the use of NSCs offers a window of opportunity for the treatment and prevention of neurodegenerative diseases, including AD, PD, HD, among others. On the other hand, the drugs approved by the Food and Drug Administration and European Medicines Agency for these types of diseases have been exclusively focused on the symptomatology, not promoting repair and reversion of these disorders. Further, several preclinical studies have already provided proof of concept that stem cell transplantation, including NSC transplantation, is beneficial in the context of neurodegenerative diseases [161,162,163,164,165], enabling the recovery of brain functions that have been lost throughout these disorders.

In addition, it is currently believed that the beneficial effects of stem cells on the injured tissues relies not only on their ability to differentiate, but also on several molecules they secrete, which in turn has been the subject of great interest. In this regard, several molecules have been already identified as part of the NSC secretome, namely, neurotrophic factors, neuroregulatory molecules, microRNAs and exosomes. The paracrine hypothesis is therefore based on the fact that NSCs act as a natural source of bioactive molecules, which play extremely important roles in the CNS tissue. For example, and as we previous stated, many of these molecules are involved in processes of cell repair, proliferation and neural differentiation.

Emerging evidence has also revealed that NSC-derived secretomes have an important antioxidant role, rescuing recipient cells from oxidative damage. Indeed, and as previously mentioned, neurotrophic factors secreted by NSCs might also act in antioxidant cellular defenses of target neural cells. For example, neurotrophic factors can either promote the expression and activity of antioxidant enzymes, allowing a direct scavenge of free radicals; or directly act on mitochondria, enhancing their functioning or preventing their impairment. Notably, recent findings have also suggested that EVs delivered by NSCs are a source of proteins involved in cellular protection against oxidative stress and, importantly, also carry functional mitochondria to rescue respiration and cell survival in recipient cells [133]. Surely, this is an interesting point to further explore in the future, once it represents an innovative mechanism to improve endogenous NSC protection throughout life. This, in turn, might be also stimulated through non-invasive metabolic interventions, such as physical exercise, dietary habits and other environmental factors that positively affect the host bioenergetic metabolism.

On the other hand, in the last decades, there has been a growing interest in new combinations and exogenous supplementations of antioxidants to amplify the antioxidant defense and prevent harmful CNS disorders. In fact, neurodegenerative diseases are closely associated with oxidative stress and mitochondrial dysfunction, supporting the idea that an enhancement of the antioxidant response might have effective implications for the progression of neurodegeneration processes associated with these types of diseases. Of note, mitochondria take part in both regeneration and degeneration processes, and despite being the main source of cellular ROS, they are also well equipped with antioxidant defense systems. For this reason, the efficiency of exogenous antioxidant therapies has become increasingly doubtful, especially due to the fact that the low levels of ROS are key signaling molecules for a wide range of cellular events, including neural differentiation [48].

Finally, despite being pivotal in modulating NSC fate, mitochondria are also crucial in regulating the cell secretome [166,167]. Although there is some information regarding the role of mitochondria in the regulation of cell and stem cell secretome, very little is known about the influence of the mitochondrial metabolism in endogenous NSCs secretome. Thus, additional studies are eagerly needed to provide a better understanding of the role of mitochondrial metabolism and antioxidant systems in NSC-based neuroplasticity. This will certainly translate into important knowledge and novel pharmacological strategies to improve endogenous NSC protection in the elderly and enhance the life quality of people with neurological disorders.

## Figures and Tables

**Figure 1 antioxidants-10-01088-f001:**
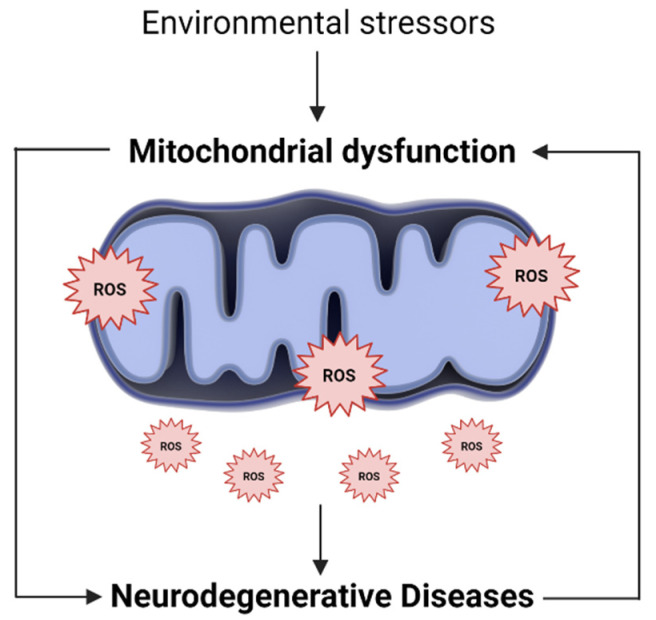
Oxidative stress and mitochondrial dysfunction loop. Along with endogenous sources, several environmental and exogenous stressors may lead to an excessive generation of mitochondrial ROS. The higher ROS levels, in turn, promote mitochondria injury, alter their metabolic enzymes and lead to its further dysfunction. In this way, mitochondrial oxidative stress can be the cause and consequence of mitochondrial dysfunction, being involved in the pathogenesis of several neurodegenerative diseases.

**Figure 2 antioxidants-10-01088-f002:**
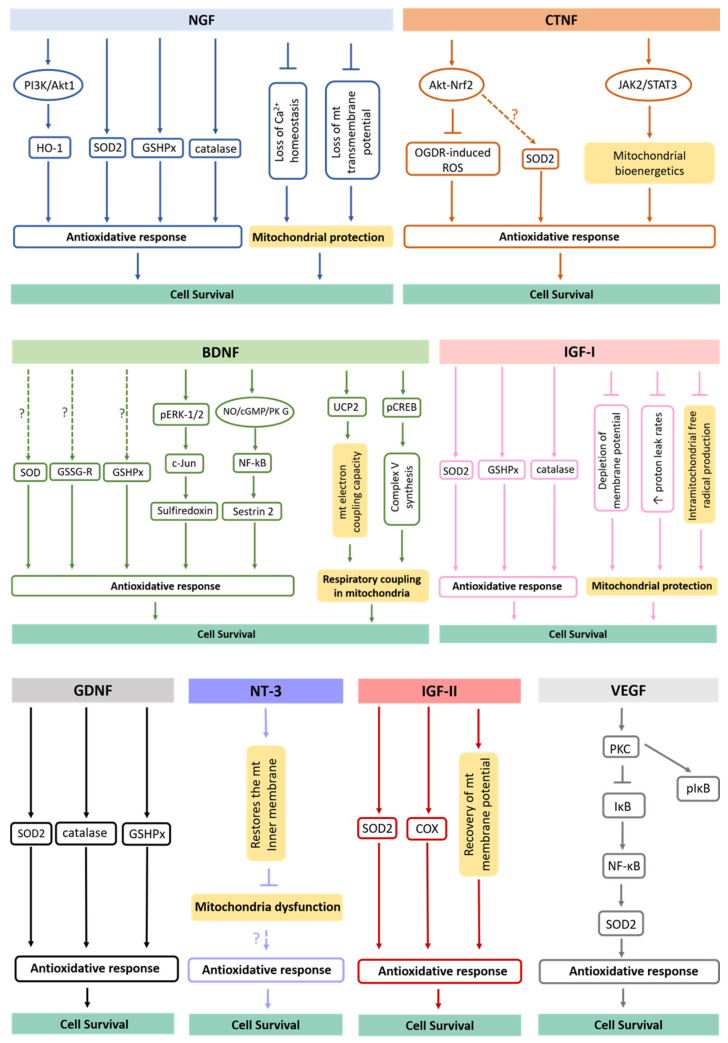
Antioxidative mechanisms triggered by secreted neurotrophic factors. It has been shown that neurotrophic factors act in major antioxidant defense pathways, rescuing target cells from oxidative damage. The underlying mechanisms of this process involve upregulation of antioxidative proteins/enzymes, increase of their activation levels, alteration in Ca^2+^ homeostasis, activation of signaling pathways, changes in mitochondrial morphology and dynamics, among others. mt., mitochondria.

**Figure 3 antioxidants-10-01088-f003:**
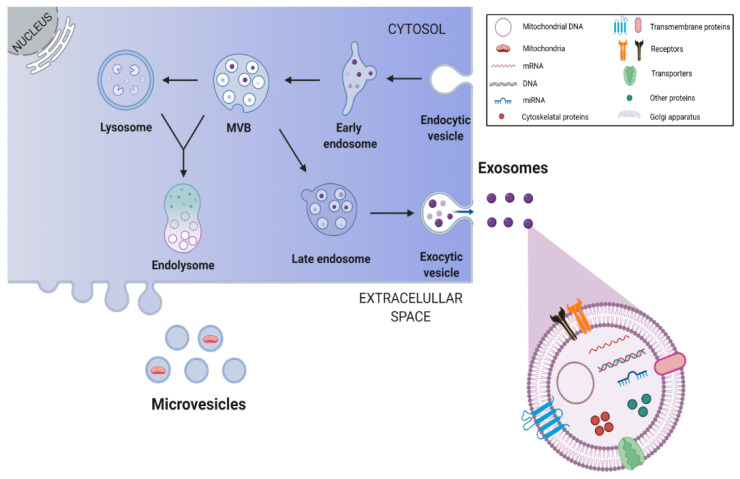
Biogenesis and release of stem cell-derived exosomes. Endosomes are produced through invaginations of the plasma membrane, which in turn generate intraluminal vesicles by budding into early endosomes and multivesicular bodies (MVBs). Through the endosomal sorting complex required for transport pathways, exosomes are formed and packaged within endosomes. Some of the MVBs fuse with lysosomes to form the endolysosome, where all components are digested, while others fuse with the plasma membrane resulting in exosome release into the extracellular space. This image was created with BioRender.com.

**Figure 4 antioxidants-10-01088-f004:**
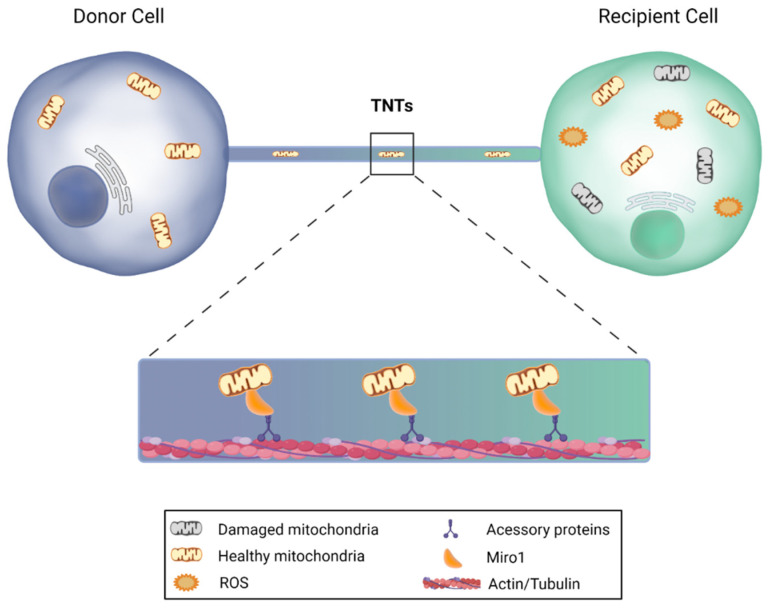
Mitochondrial transfer via tunneling nanotubes. Under certain circumstances, cells are capable of sending mitochondria to recipient cells through nanotubular actin-based structures. The donor cell extends a filopodia-like protrusion toward the recipient cell, forming a strait connected bridge between cells, in which the transfer is feasible in both directions. Miro1, a protein that is anchored to the outer mitochondrial membrane, binds to the motor proteins kinesin and dynein, and promotes mitochondrial transfer, leading to the restoration of aerobic respiration and improvement of target cell survival. This image was created with BioRender.com.

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
