# Peer review of "Oxidative-Signaling in Neural Stem Cell-Mediated Plasticity: Implications for Neurodegenerative Diseases"

_antioxidants, 2021, doi:10.3390/antiox10071088_

Round 1

Reviewer 1 Report

The review is well written and tackles most of the relevant aspects of the antioxidant properties of NSC-derived secretome and the role of mitochondria, and the relevance for neurodegenerative conditions. I have just minor points:

- Authors should revise some spelling/grammar mistakes. Some examples (not limited to):

Line 79: “once” change to “since”

Line 90: “is due” change to “is due to”

Line 186: “ we will further discussed” change to “we will further discuss”

Line 212: “it thus appear” change to “it thus appears”

- Line 221: the authors state that “One of the main causes for the involvement of PS in FAD is the association of PSs with intracellular calcium stores of the endoplasmic reticulum (ER)…”. Alzheimer’s disease (AD) is a complex disease and many mechanisms may be operating. PS carrying FAD mutations affects other important aspects of AD, such as the length of abeta peptide and thus its propensity to form aggregates. It is true that PS carrying FAD mutations has been shown to affect Calcium stores, but the causality of such alterations has not been established. Then, I would propose to change this sentence by “One of the possible causes for the involvement of PS in FAD is…”.

Author Response

Response to Reviewer: 1

(Santos et al.; Antioxidants-1255618)

We are pleased that the Reviewer considered our manuscript well written, tacking most of the relevant aspects of the antioxidant properties of NSC-driven plasticity under neurodegenerative conditions. However, the Reviewer recommended minor revisions that we have addressed in the revised version of the manuscript. By doing so, we feel that we are submitting an improved version of our manuscript that we hope is deemed acceptable for publication in Antioxidants.

  1. Following the Reviewer’s suggestion, we have carefully revised the spelling and grammar mistakes in the article, including those mentioned in previous lines 79, 90, 186 and 212. By doing so, we hope to present a clearer and well-written manuscript.

  1. The Reviewer raised an important aspect regarding the association of presenilin with intracellular calcium stores in APP mutations occur in familial AD (FAD). For an easier understanding, we re-worded the statement on previous line 212, to “One of the possible causes for the involvement of PS in FAD is…”.

Reviewer 2 Report

In this review entitled "Oxidative-signaling in neural stem cell secretome: implications for neurodegenerative diseases" the role of oxidative stress and of the mitochondria in neurodegenerative diseases, mainly Parkinson's and Alzheimer's, is extensively described and also the relationship between neural stem cells and oxidative signaling and mitochondria. However, only a small part of the work is focused on the secretome and within this a small part on the secretome related to oxidative signaling. Therefore, the subject of the review is broader than the title indicates. In addition to this main problem, which at least implies matching the title to the content, there are some parts that are not clearly explained. Some of them would be, for example:

  1. On page 4, line 157, it says "these modifications" but no specific modifications have been mentioned previously, if not in general, of "negatively affected" molecules.
  2. The next paragraph on reasons why the brain is more susceptible to oxidative stress also does not add anything, it only lists different items, but without giving reasons or explanations.
  3. On line 179, when it says "in recent years" it does not seem correct to me because it is something that has been studied and known for quite some years, at least since the 90s.
  4. Figures 1 and 3 are very simple and do not provide much information, I think they can be deleted or with more details, more informative.
  5. In line 218, if it is not explained better, it is not understood what he means by "mitochondria dinamic events".
  6. It is also not explicitly understood why the decrease in calcium in ER affects ATP production, in line 225.
  7. In line 246, why does blocking the NMDA receptor increase glutamate in the synaptic space? If this is correct, the relationship must be explained.
  8. In the same way, also in line 283, authors have to link the first sentence of that paragraph with what is said below, explain it in greater detail, since if not, the relationship between the mutation in the SOD and the beneficial effect of NSCs is not clear.
  9. In line 598, the proteomic content of the secretome must be more detailed than other parts of the review, as it is closer to the subject of the work.
  10. The conclusions reiterate the fact that in the review, oxidative signaling in the NSC secretome is only present in a small part of the review and most reviews the role of oxidative stress and mitochondria in neurodegeneration and the role of the NSCs on this, so the conclusions are too general.

Minor points

-Mitochondria organelle? line 241

-Define PGC-α1, line 291

Author Response

Response to Reviewer: 2

(Santos et al.; Antioxidants-1255618)

We are thankful for the thorough review of the manuscript and are pleased that the Reviewer considered our study interesting and fitting within the scope of Antioxidants, giving us the opportunity to improve certain aspects of the manuscript.  The Reviewer raised a number of questions and several concerns that we have minutely addressed in a revised version of the manuscript.  By doing so, we feel that we are submitting an improved version that we hope fulfills the standard requirements for publication in Antioxidants.

  1. As the reviewer pointed out, only a small part of the manuscript focus on the secretome of neural stem cells. This is due to the fact that the influence of the mitochondria on endogenous NSCs secretome is still largely unexploited. We agree with the Reviewer in adapting the article title to the content. Thus, we have changed it from “Oxidative-signaling in neural stem cell secretome: implications for neurodegenerative diseases” to “Oxidative-signaling in neural stem cell-mediated plasticity: implications for neurodegenerative diseases”.

  1. In fact, the statement of “these modifications” was to mention the negative effects mediated by ROS in macromolecules, such as membranes, lipids, proteins, lipoproteins, and DNA. To make ourselves clearer, we re-wrote this statement.

  1. We agree with the Reviewer regarding the lack of relevant information in paragraph on the reasons why the brain is more susceptible to oxidative damage. Changes were performed in this part of the article to carefully explain our idea.

  1. The Reviewer is absolutely correct in this criticism and we apologize for this lapse. We have now reorganized the sentence to “The role of mitochondrial dysfunction and excessive levels of ROS in neurodegeneration process has been largely described throughout the last years”.

  1. The Reviewer raised concerns regarding the lack of novelty of Figures 1 and 3. We understand the criticism. However, we do believe that Figure 1, although being very simple and not novel, exhibits an important background statement for the understanding of the oxidative-signaling relevance in NSC-mediated neuroplasticity, a crucial idea throughout the article. Regarding Figure 3, we think that while elucidating the biogenesis and release of stem cell-derived exosomes, it also includes novel findings on the mechanisms by which the release of exosomes might contribute to prevent mitochondrial dysfunction in recipient cells, such as the transport of mitochondria organelles in microvesicles. For these reasons, we would like to maintain both Figures 1 and 3.

  1. For a better understanding in previous line 218, we have now added “by interfering with mitochondrial fission and fusion dynamics”.

  1. In previous line 225, we clarified the fact why the decrease in ER-calcium might affect ATP production in FAD-PS2 mutants. In fact, it is well established that calcium regulates cellular bioenergetics by either allosterically activating key metabolic enzymes and metabolite shuttles or indirectly by modulating signaling cascades. For example, the correct transport of Ca2+ between ER and mitochondria compartment enhances Kreb Cycle and sustains ATP production. Thus, considering this effect on ER-calcium content, the mutants dampen mitochondrial calcium rises, impairing ATP production.

  1. In previous line 246, we stated that NMDA receptor antagonism increases glutamate in the synaptic space. This casual-effect mechanism is now further explained in the revised version of the manuscript, where we explained that inhibition of NMDA receptor normally leads to an increase of presynaptic glutamate release.

  1. We appreciate the Reviewer’s suggestion on previous line 283. Therefore, we have reorganized the paragraph, including new information, to better explain the relationship between the mutation in the SOD and the beneficial effect of NSCs.

  1. In the previous line 598, the proteomic content of the NSC-derived secretome was specified in more detail.

  1. The Reviewer is correct in this criticism. In fact, we agree that the oxidative signalling in the NSC secretome is only present in a small part of the review and most of the available information focuses the role of oxidative stress and mitochondria in neurodegeneration and NSC-mediated plasticity. Thus, in addition to adjust the title of the manuscript we also altered the “Conclusions” to “Summary” to comport a more general conclusion, that we also tried to extend.

Minor points:

-“Mitochondria organelle” was removed.

- The peroxisome proliferator-activated receptor γ coativator 1α (PGC1α) was previously defined in line 139.

Reviewer 3 Report

This is an excellent review for oxidative signaling in neural stem cells. This reviewer judged that this review is very informative and valuable for the readers of "antioxidants". 

Author Response

Response to Reviewer: 3

(Santos et al.; Antioxidants-1255618)

We appreciate the thorough critique of the manuscript by the Reviewer and are delighted that he/she considered it an excellent review for oxidative signaling in neural stem cells with adding valuable for the readers of "Antioxidants". 

Round 2

Reviewer 2 Report

The authors have rightly reviewed and answered the questions raised. The review is more coherent and consistent now.

I just have a few additional comments on your modifications.

-In question 8, line 266, as in 7, line 241, the references that correspond to the clarified sentences should be added.
-As for the proteome analysis, in line 616, the sentence "revealed promising in neurodegenerative contexts" must be either justified or eliminated, if there is no clear relationship between the aforementioned proteins and their benefit in neurodegeneration.
-In the sentence added in the summary and future directions, line 678, the reference should also be added.

Author Response

Response to Reviewer: 2

(Santos et al.; Antioxidants-1255618)

We are pleased that the Reviewer appreciated our responses to the questions raised. However, the Reviewer recommended minor revisions that we have addressed in the revised version of the manuscript. By doing so, we feel that we are submitting an improved version of our manuscript that we hope is deemed acceptable for publication in Antioxidants.

  1. Following the Reviewer’s suggestion, we have added the respective references to the new clarified sentences, including in the Summary section. In addition, we have clarified the relevance of certain proteomic hits in the context of neurodegenerative disorders.

  1. Importantly, since we have suddenly found a recent report describing the traffic of functional mitochondria from neural stem cell-derived extracellular vesiculas, we decided to share this novel discovery in the article and readjust our discussion.
